# Entropy Production Using Ecological and Physiological Models of Stand Growth Dynamics as an Example

Victor Ivanovich Lisitsyn * and Nikolai Nikolatvich Matveev

Department of General and Applied Physics, Voronezh State University of Forestry and Technologies, 8 Timiryazeva Street, 394087 Voronezh, Russia
* Correspondence: viktor-lisicyn@yandex.ru; Tel.: +7-910-3417858

**Abstract:** According to the Prigogine–Glansdorff principle, in a thermodynamic system where non-equilibrium processes occur, the change in entropy production is negative or equal to zero. Forest plantations are an example of such thermodynamic systems. Based on the ecological–physiological models of the forest stand growth dynamics developed by the authors of this work, the specific entropy production in the growth of stands was calculated, which is known to be proportional to the entropy production. It is shown that at the age when the minimum value of the specific entropy production is reached, the biomass of the stand has a maximum value. This conclusion is an important predictive factor for practical forestry, since determining the time of the minimum entropy production allows us to determine the time interval at which the biomass of the stand reaches its maximum value.

**Keywords:** entropy; forest ecosystem; growth dynamics; ecological and physiological models; biomass

## 1. Introduction

As early as 1886, Boltzmann noted [1] that " ... the general struggle for existence of living beings is not a struggle for raw materials: for organisms it is air, water, and soil, available in abundance, and not a struggle for energy, which exists in abundance in any body in the form of heat, but a struggle for entropy, which becomes available when energy passes from the hot sun to the cold Earth"; however, as noted in [2], the main problem in the use of thermodynamic laws in ecology is that thermodynamic properties can only be calculated, but not measured, although thermodynamics provides an opportunity to describe almost all processes of the Earth system in purely thermodynamic terms. This treats the biosphere as consisting of a series of self-organizing subsystems, each optimizing entropy production within the constraints of time, feedback, and system constraints (ecosystem hierarchy).

In this work, we do not consider various approaches to describing the state of open systems far from the state of thermodynamic equilibrium [2–4], in which generalizations related to the second law of thermodynamics have been proposed, but limit ourselves to applying the Prigogine–Glansdorff principle [5], which states " ... that the evolution of such systems goes towards states with the lowest entropy production (per unit mass)".

It should be noted that entropy production is determined not only for heat fluxes and temperature gradients, as considered in physics, but also for a very wide range of ecological problems—from studying the entropy balance for individual organisms, including humans [6], to entropy production in such ecosystems as lakes [7], marshes, and forested areas [8].

The forest ecosystem is a typical example of an open thermodynamic system. The aim of this work is to show that, using the ecological–physiological models of the forest stand growth dynamics used in modern forestry, it is possible to obtain an analytical expression for the specific entropy production as a function of time, which has a minimum at the point in time at which the biomass of a stand reaches its maximum value. In contrast to a number of works in which empirical data on heat fluxes, radiation energy, and energy of chemical

reactions were used to calculate entropy production, in the present work the analytical expression for specific entropy production was obtained based on phenomenological models in which the parameters of ecological and physiological models developed earlier by the authors of the article [9,10] were used.

## 2. Material and Methods

The following processes can be realized in open thermodynamic systems: (1) absorption of short-wave solar radiation—the process of differentiation; and (2) the process of overall biomass growth, which is associated with resource consumption due to the costs of respiration and competition. As a result of these processes, a negative entropy flux enters the system, while positive entropy is produced in the system itself. During the growth of the stand over time, the biomass reaches the maximum value, which corresponds to the stationary state in the ecosystem [9,10]. As the age of the stand increases further, the stationary state of an open thermodynamic system transitions to a state in which the biomass decreases and the entropy of the system increases (ecosystem decay occurs).

It is well known (see, for example [4]) that the entropy change for an open system dS consists of an external exogenous contribution from the environment $d_e S = \left( q_{in}^S - q_{aut}^s \right)$ (where $q_{in}^S$ and $q_{aut}^s$ are internal and external entropy inflows and outflows) and $d_i S$ is an internal endogenous contribution due to processes inside the system, which must always be positive in accordance with the second law of thermodynamics.

$$dS = d_e S + d_i S \tag{1}$$

Three situations are possible for the balance of entropy.

$$\frac{dS}{dt} = \frac{d_e S}{dt} + \frac{d_i S}{dt} > 0 \tag{2a}$$

$$\frac{dS}{dt} = \frac{d_e S}{dt} + \frac{d_i S}{dt} < 0 \tag{2b}$$

$$\frac{dS}{dt} = \frac{d_e S}{dt} + \frac{d_i S}{dt} = 0 \tag{2c}$$

In the first case (2a), the system, in accordance with the second law of thermodynamics, tends to an equilibrium state in which all thermodynamic processes cease. In the case of (2b), the entropy flux is negative and ordering processes may proceed. This is possible under the condition $q_{in}^S < q_{aut}^S$, i.e., the creation of order in the system must be associated with a larger entropy flux out of the system, rather than into the system. This means that the system must be open, or at least non-isolated. Case (2c) corresponds to a stationary situation, for the description of which the equations for the balance of energy (U) and entropy (S) are used.

$$\frac{d_e U}{dt} = -\frac{d_i U}{dt}, \ \frac{d_e S}{dt} = -\frac{d_i S}{dt} \tag{3}$$

According to Prigogine's theorem, the directional criterion of irreversible processes in an open system near the equilibrium position, in the case of processes characterized by constant flows of physical quantities occurring in it, is as follows: as it approaches a stationary state, the entropy production rate within the open system monotonically decreases, gradually approaching its minimal positive constant value (here, we follow the ideas proposed in [4] by S. E. Jorgensen and Yu. V. Svirezhev).

When a system approaches a stationary state, forces and flows change in such a way that entropy production is continuously decreasing [11]. Prigogine's theorem can be considered as a criterion of system evolution in the linear region. The directional criterion of irreversible processes in an open system in the nonlinear region was formulated by Glansdorff and Prigogine [5] (for biological systems it was considered in [12]). In this case, the change in entropy production caused by fluctuations in the generalized forces is either negative or negative/equal to zero (hereinafter, we are interested in the nature of the

entropy production dependence in the region of such values of time when the system is near the stationary state, far from the equilibrium position).

Let us carry out further consideration for a forest stand with biomass $M$. Theoretically, we should expect that two processes make the main contribution to the entropy change: the total biomass growth and cell division. In order to separate these effects, let us consider the change in specific entropy—entropy per biomass unit $\sigma = S/M$.

$$\frac{d}{dt}\left(\frac{S}{M}\right) = \frac{1}{M}\frac{dS}{dt} - \frac{1}{M}\left(\frac{S}{M}\right)\frac{dM}{dt} \text{ or } \frac{dS}{dt} = M\frac{d\sigma}{dt} + \sigma\frac{dM}{dt} \tag{4}$$

The process of differentiation leads to a decrease in the specific entropy, as the order in the system increases, while the growth of biomass corresponds to the positivity of the derivative $dM/dt$; therefore, the change in the entropy of an organism is determined by a combination of the negative (differentiation) and positive (growth) derivatives [4].

## 3. Results

Equations (2a–c) can be rewritten in a more general form. To do this, we present the expression for entropy as S = σM = ρσV, where ρ is the biomass density, and V is the volume of the forest stand. This expression does not contain a statement about the entropy structure. Here, we use the value σ—entropy density. Suppose that

$$\frac{d_i S}{dt} = \alpha M \text{ and } \frac{d_e S}{dt} = bF \tag{5}$$

where $\alpha$ is a function of time, b is a constant whose value is determined in the ecological and physiological model [9,10], and F is the surface area of the stand.

It is worth emphasizing that the entropy production within the system is not proportional to the biomass of the stand, but it is related by a functional relationship (the first formula in Equation (5)). The validity of phenomenological models (5) can be verified by calculating entropy production at the point in time when the forest ecosystem reaches a stationary regime in which entropy production is minimal. If the model calculation confirms the entropy production value is minimal, then, at this stage of model building, we can make a judgement about the adequacy of the presented model. Equation (5) were used in [4] to prove the allometric principle, which is used in practical forestry.

To calculate the entropy production during stand growth, we start by considering Equation (2c) of this paper. Then we have:

$$\frac{dS}{dt} = \frac{d_i S}{dt} - \left|\frac{d_e S}{dt}\right| = M\frac{d\sigma}{dt} + \sigma\frac{dM}{dt} \tag{6}$$

The value $\alpha = \left(\frac{1}{M}\right)\frac{d_i S}{dt}$ has the meaning of the specific entropy production, for which, according to the Prigogine–Glansdorff [12] principle:

$$\frac{d\alpha}{dt} \leq 0 \tag{7}$$

The area and the biomass are related by a relationship of the general form [4]

$$\Phi(F,M) = 0 \tag{8,}$$

the analytical expression of which is given by the ecological–physiological model of the forest stand growth dynamics. Currently, a sufficiently large number of ecological–physiological models are known. The classification of ecological–physiological models is rather conventional, but, in general, they can be divided into analytical [13], simulation [14,15], and analytical–simulation [16–18]. We have referred here to those papers in which this division is most pronounced (another classification is also possible, which does not change in principle the meaning of forest ecosystem modeling [14]). Note that the importance

of thermodynamic laws for substantiation of ecological and physiological models of the forest stands growth dynamics was first considered in the works of G. A. Aleksandrov and G. S. Golitsyn [19].

Below, we present the main ideas and provisions of ecological and physiological models, a detailed description of which was published in our papers [9,10]. We based our model [9] on the fact well-known in forestry that when the biomass of a stand reaches a certain age, it reaches its maximum, after which the biomass begins to gradually decrease (this dependence is typical of many stands (see, for example, [13,20])).

The presence of such a maximum indicates that the system has reached a state where $\frac{dM}{dt} = 0$ at $t = t_{max}$.

If condition (2c) is satisfied, there is entropy balance in the ecosystem and it is in a stationary state [4]. In [9] it is shown that at the moment of time when the plantation biomass has a maximum value the forest ecosystem is in a stationary state; then, the specific entropy production reaches the minimum positive value. This conclusion is in full agreement with the Prigogine–Glansdorff principle [12].

Note that the proof of Prigogine's theorem (see, e.g., [4]) uses linear laws and Onzager's reciprocity relation, i.e., Prigogine's theorem is valid for linear irreversible processes with invariable external parameters; however, this statement is not a necessary condition for the appearance of a stationary state. If the change in external parameters is slower than the change in the system parameters, then the stationary state of the system may occur. If the change in external parameters is slower than the change in system parameters, then a stationary state of the system is possible, which, as indicated above, is observed at time $t = tmax$.

The objects of modeling are the plantation biomass $m$, and the number of trees $N$ per hectare of plantation. They are related by the obvious correlation:

$$M = mN \tag{9}$$

The equations for the number of trees $N$ and the biomass of a single-tree $m$ stand are as follows: [9]

$$m(t) = m_\infty (1 + \beta exp(-at))^p \tag{10}$$

$$N(t) = N_0 \frac{(1 + \beta_1)^{p_1}}{(1 + \beta_1 exp(-a_1 t))^{p_1}} \tag{11}$$

Equations (10) and (11) are solutions of the following system of differential equations, respectively:

$$\frac{dm}{dt} = fgm^q - rm \tag{12}$$

$$\frac{dN(t)}{dt} = \left(-f_1 g_1 m^{q_1 - 1} + r_1\right) N(t) \tag{13}$$

where $f$ is the specific resource uptake rate; $r$ is the specific resource consumption rate for tree biomass; $f_1$ and $r_1$ are the same values modified for the number of trees per hectare; and $\beta$, $\beta_1$, $a$, $a_1$, $p$, and $p_1$ are model parameters. Equation (12) is the well-known Bertalanffy equation [21], and Equation (13) is presented in [9]. In [13] it is written that the absorbing surface area F and biomass of plantation under the condition of canopy closure are related by the relation $F = gM^q$, where $g$ and $q$ are allometric parameters. In our ecological and physiological stand model [9], the stand biomass is defined by Equation (9), and the biomass of one tree and the number of trees per hectare are expressed by Equations (10) and (11), respectively, and, for them, the parameters $q$ and $q_1$ are different, so the stand surface area and biomass are related by the following relation:

$$F = gm^q N^{q_1} \tag{14}$$

where $q$ and $q_1$ are the names of allometric parameters of the model, $m$ is the average value of the biomass of an individual tree, and $N$ is the number of trees per 1 ha. For this, we

substitute Equation (5) into Equation (6). After transferring the term "*-bF*" to the right side and dividing by the biomass of the plantation *M* taking into account expression (14), we have the following relation for function *α(t)*:

$$\alpha(t) = \frac{d\sigma}{dt} + \frac{\sigma}{M}\frac{dM}{dt} + bg\frac{m^q N^{q_1}}{M} \tag{15}$$

Let us define the function:

$$\delta_\sigma(t) = \alpha(t) - \frac{d\sigma}{dt} \tag{16}$$

It will have the meaning of effective specific entropy production. Let us consider the specific entropy σ as a function parameter $\delta_\sigma(t)$. We know that at time $t = t_{max}$, at which the stand biomass reaches a maximum value, the forest ecosystem is in a stationary state, i.e., at time $t_{st} = t_{max}$, and the specific entropy according to the Prigogine principle is minimal, then the derivative *dσ/dt* = 0. At this time point, the values of the effective specific entropy $\delta_\sigma(t_{st})$ and the specific entropy $\alpha(t_{st})$ are equal, which follows from Equation (16), and, since we are interested in the time interval near $t_{st}$, and the functions $\delta_\sigma(t)$ and $\alpha_\sigma(t)$ are monotonic, their differences in this small interval will be insignificant, and at the stationarity point itself they are identically equal. One of the goals of our work was, first of all, to calculate the value of production of specific entropy near the stationary point, where this value should reach a minimum value according to the Prigogine principle. The time $t_{st} = t_{max}$ for forest ecosystems is usually more than a hundred years, which is a sufficiently large value in terms of stand age. As follows from [22], at large values of time of open systems fairness of Prigogine's principle does not cause doubts, in contrast to small times of ecosystems development, where it is preferable to use MEPP principle. If our proposed model for calculating entropy production corresponded to reality, then we should have a minimum value of specific entropy production from the stationary point. Of course, the specific entropy itself is a function of time, which is what our model is based on, with, of course, the value of the specific entropy itself being interesting. The derivative of total biomass has the form:

$$\frac{dM}{dt} = \frac{dm}{dt}N + \frac{dN}{dt}m \tag{17}$$

Thus, we obtain all the analytical expressions of the functions necessary to calculate the effective specific entropy production—*M*, *m*, *N*, $\frac{dm}{dt}$, and $\frac{dN}{dt}$. It is not difficult to see that the effective specific entropy production $\delta_\sigma(t)$ is determined by the following expression:

$$\delta_\sigma(t) = \frac{\sigma}{M}\frac{dM}{dt} + bg\frac{m^q N^{q_1}}{M}, \tag{18}$$

i.e., it depends on the entropy density σ as a parameter. Now, we see the meaning of the introduction of the value of effective specific entropy production. We know neither the form of the function σ as a function of time, nor its derivative. Since all model parameters and functions *M*, *m*, *N*, $\frac{dm}{dt}$, and $\frac{dN}{dt}$ are known to us, then, in accordance with the Prigogine–Glansdorff principle [12], the minimum positive value $\alpha(t_{st})$ when varying $\sigma$ will give us both the value of the entropy density and the type of dependence of the effective specific entropy production on time near the stationary point $t_{st}$.

## 4. Discussion

All initial numerical values were taken from the authors' works [9,10]. To calculate the functions $\delta_\sigma(t)$ and $\alpha_\sigma(t)$, programs for computers were compiled in the PTC Mathcad language environment—Entropy Production Calculation Program for Single-Tree Stand [23].

The results of calculating the effective specific production as a function of time are shown in Figure 1.

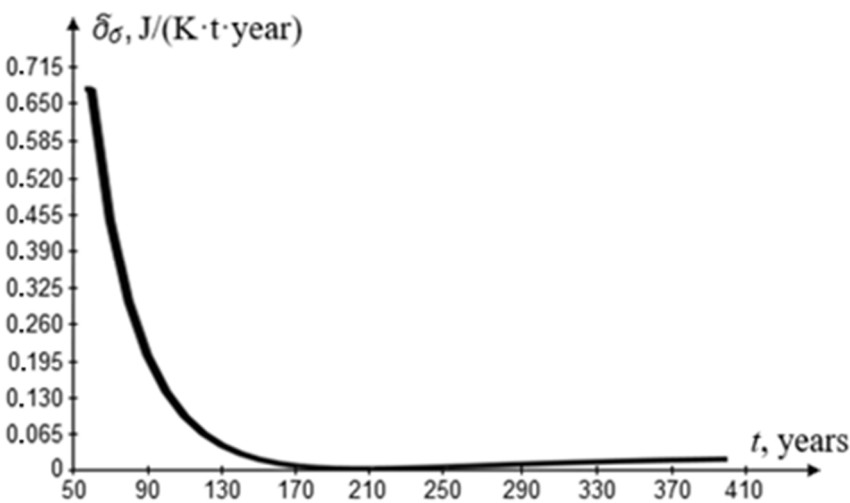

**Figure 1.** Dependence of the effective specific entropy production on time for a single-tree stand. The Y-axis shows the effective specific entropy production δ_σ (t) in J/(K t year). The X-axis shows time in years.

The minimum value of the effective specific entropy production is reached at $t_{st}$ = 200 years, which is consistent with the maximum value of the total biomass [20]. The value of the parameter σ—the specific entropy density—in this case, is σ = 2 J/(K t). If the specific entropy density rises above two (by 0.1%), the specific entropy production in the vicinity of the stationary point becomes negative, which contradicts the second law of thermodynamics. If the value of the specific entropy density is less than two, the minimum is undetectable over the entire time interval, which is inconsistent with the Prigogine–Glansdorff principle. As stated in [2], "The main problem in transferring the laws of thermodynamics to ecology is that almost all thermodynamic properties can only be calculated, but not measured". The obtained theoretical values of the specific entropy production and the specific entropy density are very difficult to compare with the data obtained by other authors, since in most cases are given either relative values of entropy production in relation to the maximum entropy value of the short-wave solar radiation, or per unit area of spruce forest stand. The corresponding conversion to the same units may not be correct; however, in order of magnitude, the entropy production values presented by us and in [8,24] are quite comparable; in addition, it is not possible to find the same objects of research, and at the same time the approaches to determining the methods of calculating the production of entropy are fundamentally different.

The principles of constructing an ecological–physiological model of a mixed stand based on the corresponding model of a homogeneous stand are considered in the authors' work in [10] (it should be noted that, despite the large number of works (see, for example, [18]) devoted to the construction of mixed-stand models, we are not aware of calculations by other authors for ecological-physiological models using data on real mixed forests).

To calculate specific entropy production during the growth of mixed stands, formulas similar to those for single-tree stands are used. We do not present their analytical expressions, since they are rather cumbersome and do not differ in principle from the above Equations (15)–(18).

The results of the calculations are shown in Figure 2.

The minimum value is reached at t = 190 years, which is in agreement with the maximum value of the total biomass [20] calculated using the model we developed [10]. The value of the specific entropy density is 7.34 J/(K-t).

Verification was performed in the case of a single-species forest stand for full (normal) pine stands of 1b bonitet, and, in the case of a two-species forest stand, for full two-tree

aspen–spruce stands of bonitet 1. The corresponding empirical values of the plantation biomass and the number of trees per hectare were taken from [20].

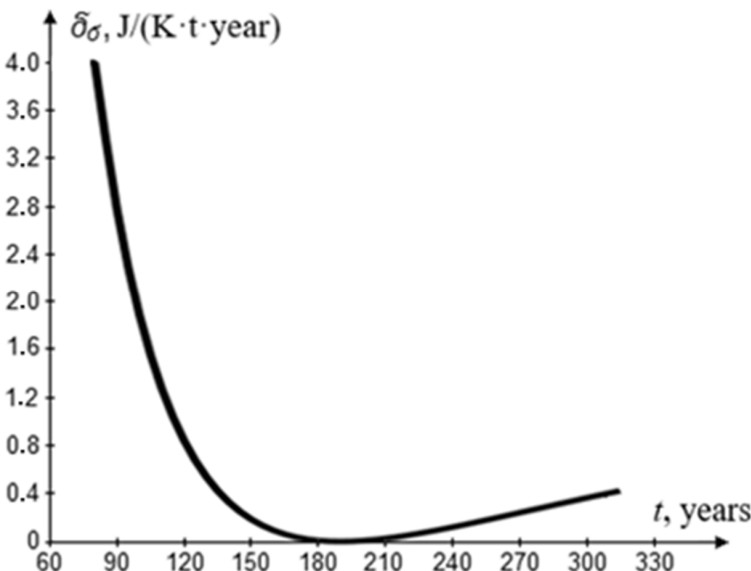

**Figure 2.** Dependence of effective specific entropy production on time for a two-species forest stand. The Y-axis shows the values of effective specific entropy production in J/(K t year). On the X-axis is time in years.

## 5. Conclusions

The results of this study suggest that the proposed approach to calculating the specific entropy production and the specific entropy density and the approaches developed in calculating entropy balances will yield similar qualitative and quantitative results. It is very important from our point of view to apply our proposed approach to problems related to the principle of maximum entropy production (MEPP), which states that thermodynamic processes far from thermodynamic equilibrium adapt to steady states in which they dissipate energy and produce entropy at the maximum possible rate. It is well known that application of MEPP at the ecosystem scale results in maximum stand productivity.

As indicated above, we have made an assumption about the relationship between the principle of minimum production of entropy as per Prigogine–Glansdorff [12] and the principle of MEPP. The existence of such a relationship is discussed in [22].

The construction of a formalized model according to the principles of systems analysis is a step-by-step procedure, in which the first steps in model construction are not free from drawbacks, and even more so for models that use the thermodynamic approach to describe the properties of nonequilibrium ecological systems [2]. To these shortcomings we include:

1. The paper actually presents two models. One is an entropy production model based on the introduction of effective entropy density production. The second is an ecological–physiological model of the dynamics of stand growth. It is difficult to present them separately in this paper, since they are interrelated. This circumstance makes it difficult to present the proposed approach clearly;
2. The parameters of the ecological–physiological model depend on the choice of initial data on the biomass of the stand;
3. We know rather few reliable data about biomass of mixed plantations which influence the statistics of the presented results;
4. However, the resulted deficiencies, in our opinion, do not reduce the value or the novelty of the results presented in our work.

**Author Contributions:** Conceptualization, V.I.L. and N.N.M.; methodology, V.I.L. and N.N.M.; validation, V.I.L. and N.N.M.; formal analysis, V.I.L. and N.N.M.; investigation, V.I.L.; resources, V.I.L.; data curation, V.I.L.; writing—original draft preparation, V.I.L.; visualization, V.I.L.; and supervision, V.I.L. All authors have read and agreed to the published version of the manuscript.

**Funding:** This research received no external funding.

**Informed Consent Statement:** Informed consent was obtained from all subjects involved in the study.

**Data Availability Statement:** The necessary data are taken from open sources. Reference [20] from the list of references—Shvidenko A.Z., Schepaschenko D.G., Nielson S., Buluys Y.I. Tables and models of growth and biological productivity of plantations of major forest forming species of Northern Eurasia (regulatory and reference materials); 2nd edition; Rosleskhoz, International Institute of Applied Systems Analysis: Moscow, Russia, 2008.

**Conflicts of Interest:** The authors declare no conflict of interest. The funders had no role in the design of the study; in the collection, analyses, or interpretation of data; in the writing of the manuscript; or in the decision to publish the results.

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
