# Peer review of "Entropy Production Using Ecological and Physiological Models of Stand Growth Dynamics as an Example"

_forests, doi:10.3390/f13111948_

Round 1

Reviewer 1 Report

The paper differs from the mainstream of modern forestry literature in a very interesting manner.

This reviewer has some doubt whether the sign convention of Eq. (15) is in concert with that of Eq. (5).

The formulation of Eq. (16) might require some further clarification.

It is not clear to this reviewer where the numerical values used in the production of Figs. 1 and 2 arise. Is the specific entropy density taken as a fitted constant, instead of variable with time?

Apparently strong statements are given in Conclusions, regarding the principle of maximum entropy production (MEP). Such statements possibly should be backed up with extensive referencing, along with a more detailed discussion. Such a discussion might partially appear already in the Introduction.

Applications of the MEP possibly should be explained in the Introduction, and further prospects outlined in the Discussion.

Reviewer 2 Report

1. In the last sentence of the abstract, the author should clarify why is important about the time interval at which the forest stand reaches its maximum value.

2. In the introductory section you can cite other people's literature, naming the shortcomings of it, to draw out the importance of this article

3. The model formula should be inside the methods, and what is calculated by the model should be in the results.

4.lines 221-223. This sentence is too vague, the specific comparative values should be written and clearly shown. It is best to compare the values with other models and make a table to illustrate the accuracy of the model.

5. It is necessary to write about the shortcomings of the model in the discussion

Reviewer 3 Report

Entropy Production Using Ecological and Physiological Models of Stand Growth Dynamics as an Example

The basic science of this paper is not conducted in a good way and is of inappropriate standard. The author and his team write this paper according to journal scope and modern trends but failed. I am glad to review this paper because it’s very related to my research and I found there is no sense of research. This is not article. According to my research experties. Its look like a report or mini review. Literature review of this study is insuffiient and very weak. If authors want to publish this research then they will modify this research. Moreover, they should provide some novelty or enhance the significance of the research. Furthermore, the paper is not well-structured. At this stage, I can’t provide a complete review.

Best of luck with your paper.

Round 2

Reviewer 1 Report

The idea of this reviewer within the first review round was not to ask the Authors to write a personal letter to the reviewer. The idea was that the manuscript possibly should be modified, to help a wider audience to understand the content.

It appears that the Authors have neglected the question of the sign convention, along with the other questions raised. 

In addition to the issues raised in the first round, the Authors possibly should justify why they are leaning to Eq. (2c) on line 146.

Phrasing on line 158 should be changed.

The meaning of F should be explained in more detail.

Reviewer 2 Report

I have no further comments.

Reviewer 3 Report

I can't find any major changes in this manuscript. The author should need more time to revise this manuscript as per journal criteria.

Best of luck
